# Blood Neurofilament Light Chain and Glial Fibrillary Acidic Protein as Promising Screening Biomarkers for Brain Metastases in Patients with Lung Cancer

**DOI:** 10.3390/ijms25126397

**Published:** 2024-06-10

**Authors:** Su-Hyun Kim, Beung-Chul Ahn, Dong-Eun Lee, Ki Hoon Kim, Jae-Won Hyun, Min Jeong Kim, Na Young Park, Ho Jin Kim, Youngjoo Lee

**Affiliations:** 1Department of Neurology, Research Institute and Hospital of National Cancer Center, 323 Ilsan-ro, Ilsandong-gu, Goyang 10408, Republic of Korea; 2Center for Lung Cancer, Division of Hematology and Oncology, Department of Internal Medicine, Research Institute and Hospital of National Cancer Center, Goyang 10408, Republic of Korea; 3Biostatistics Collaboration Team, Research Core Center, Research Institute, National Cancer Center, Goyang 10408, Republic of Korea

**Keywords:** brain metastases, lung cancer, biomarkers

## Abstract

The diagnosis of brain metastases (BMs) in patients with lung cancer (LC) predominantly relies on magnetic resonance imaging (MRI), a method that is constrained by high costs and limited accessibility. This study explores the potential of serum neurofilament light chain (sNfL) and serum glial fibrillary acidic protein (sGFAP) as screening biomarkers for BMs in LC patients. We conducted a retrospective analysis of 700 LC cases at the National Cancer Center, Korea, from July 2020 to June 2022, measuring sNfL and sGFAP levels at initial LC diagnosis. The likelihood of BM was evaluated using multivariate analysis and a predictive nomogram. Additionally, we prospectively monitored 177 samples from 46 LC patients initially without BM. Patients with BMs (*n*= 135) had significantly higher median sNfL (52.5 pg/mL) and sGFAP (239.2 pg/mL) levels compared to those without BMs (n = 565), with medians of 17.8 pg/mL and 141.1 pg/mL, respectively (*p* < 0.001 for both). The nomogram, incorporating age, sNfL, and sGFAP, predicted BM with an area under the curve (AUC) of 0.877 (95% CI 0.84–0.914), showing 74.8% sensitivity and 83.5% specificity. Over nine months, 93% of samples from patients without BM remained below the cutoff, while all patients developing BMs showed increased levels at detection. A nomogram incorporating age, sNfL, and sGFAP provides a valuable tool for identifying LC patients at high risk for BM, thereby enabling targeted MRI screenings and enhancing diagnostic efficiency.

## 1. Introduction

Brain metastases (BMs) are a common site of distant metastasis in lung cancer (LC), which is a significant cause of low survival rates and poor quality of life. The incidence of BM at diagnosis of non-small cell lung cancer (NSCLC) and small cell LC (SCLC) is 15–20%, with half of the cases being asymptomatic and up to 40% and 50–80% being NSCLC and SCLC, respectively, during the disease course [1,2,3]. BM number and size are crucial independent prognostic factors in LC [4]. Thus, early BM diagnosis is a prerequisite for preventing the development of neurological symptoms and effective tumor control with stereotactic radiosurgery or targeted therapies [5,6].

Magnetic resonance imaging (MRI) is the gold standard for detecting BM in patients with LC due to its superior sensitivity. Despite existing protocols recommending brain MRI at initial LC diagnosis [7,8], practical challenges such as substantial cost, the time-consuming nature of conducting and interpreting scans, and limited accessibility contribute to variable compliance with these guidelines. For example, in the UK, adherence gaps are evidenced by the fact that 32% of stage III patients do not undergo the recommended pre-treatment imaging in alignment with NICE guidelines [9]. Also, there is a lack of guidance concerning regular surveillance with brain MRI in patients who do not initially present with BM. In clinical practice, performing frequent MRI surveillance over several years in all patients with LC is not feasible. Therefore, identifying additional readily accessible biomarkers to enhance BM detection is crucial. 

Blood is the preferred choice when seeking biomarkers for cancer screening as it offers minimal invasiveness and accessible collection. Neurofilament light chain (NfL) and glial fibrillary acidic protein (GFAP) are promising candidates, with NfL indicating central nervous system axonal loss and GFAP reflecting astrocyte activation [10,11]. Several recent studies have indicated higher blood NfL or GFAP levels in patients with BMs than in those without BMs [12,13,14,15]. However, the previous studies were limited by small sample size, a lack of control for confounding factors such as age, and the potential effects of treatment on NfL and GFAP levels. In this study, we aimed to assess the usefulness of blood NfL and GFAP levels in identifying the presence of BM in a large cohort of patients during the initial staging of LC. Additionally, we explored the potential of these biomarkers as indicators for monitoring BM occurrence during treatment in a prospective cohort of patients without BM during LC diagnosis.

## 2. Results

The demographic and clinical characteristics of 700 patients with LC (135 patients with BMs and 565 without BMs) are detailed in Table 1. Among the 135 patients with BMs, 25 (18.5%) had neurological symptoms, including headaches, dizziness, seizures, or weakness. Patients with BMs (sNfL: median, 52.5 pg/mL; interquartile range [IQR], 27.6–97.8 pg/mL; sGFAP: median, 239.2 pg/mL; IQR, 167.7–414.8 pg/mL) had significantly higher sNfL and sGFAP levels than non-BM patients did (sNfL: median, 17.8 pg/mL; IQR, 13–27.5 pg/mL; sGFAP: median, 141.1 pg/mL; IQR, 105.3–184.9 pg/mL; *p* < 0.001 for both; Figure 1). Within the BM group, symptomatic patients had higher sNfL (median, 119 pg/mL; IQR, 65.1–192.9 pg/mL vs. median, 45.9 pg/mL; IQR, 23.2–68.6 pg/mL) and sGFAP levels (median, 806 pg/mL; IQR, 511.5–2104.5 pg/mL vs. median, 229.8 pg/mL; IQR, 140.2–319.7 pg/mL) than asymptomatic patients did (*p* < 0.001). We conducted multivariable logistic regression analyses to explore the potential factors associated with BM, including sNfL and sGFAP levels (Table 2). Age, along with levels of sNfL (odds ratio [OR] 6.111, 95% confidence interval [CI] 4.059−9.199) and sGFAP (OR 4.164, 95% CI 7.472−7.012), independently correlated with BM risk in patients with LC. Specifically, older age groups (60–69 years: OR, 0.402; 95% CI 0.224–0.722; >70 years: OR, 0.098; 95% CI 0.049–0.198) demonstrated a lower risk of BMs than did those under 60 years of age, highlighting age, alongside sNfL and sGFAP, as critical independent factors for assessing BM probability. The probability of BM was modeled using a logistic regression analysis, where the logit of the probability (*P*(*X*)) of BM is given by:logit *P*(*X*) = −13.9719 − 0.9112 × (Age: 60–69) − 2.3215×(Age ≥ 70) + 1.8101 × log(NfL) + 1.4264 × log(GFAP).

Consequently, the probability of BM, *P*(*X*), can be calculated as:$$P(X)=\frac{\exp(-13.9719-0.9112\times (\mathrm{Age}:\;60\text{–}69)-2.3215\times (\mathrm{Age}\geq 70)+1.8101\times \log(\mathrm{NfL})+1.4264\times \log(\mathrm{GFAP}))}{1+\exp(-13.9719-0.9112\times (\mathrm{Age}:\;60\text{–}69)-2.3215\times (\mathrm{Age}\geq 70)+1.8101\times \log(\mathrm{NfL})+1.4264\times \log(\mathrm{GFAP}))}$$

The nomogram, combining age, sGFAP, and sNfL predicted BM with a higher accuracy (AUC 0.877, 95% CI 0.84–0.914) than did the nomogram using sNfL (AUC 0.837) or sGFAP alone (AUC 0.775), showcasing the benefit of a multifactorial approach (Figure 2 and Table 3). The Hosmer–Lemeshow test showed a satisfactory fit (*p* = 0.113), indicating no significant deviation from the perfect fit within our model (Figure 2C). At a predictive probability threshold of 0.2, the nomogram achieved a sensitivity of 74.8%, a specificity of 83.5%, a positive predictive value (PPV) of 52%, a negative predictive value (NPV) of 93.3%, and an accuracy of 81.9%, optimizing clinical decision making for BM screening. Subgroup analysis in patients with advanced LC (stages III–IV, n = 379) demonstrated that age, sNfL, and sGFAP remain significant predictors of BM presence in the multivariable analysis (Appendix A); the developed nomogram for this subgroup showed excellent discrimination, with an AUC of 0.867 (95% CI 0.826–0.908) (Appendix A). At a chosen cutoff of 0.4, the model achieved a sensitivity of 72.6%, specificity of 85.3%, PPV of 73.1%, NPV of 84.9%, and overall accuracy of 80.7%.

We prospectively monitored 177 samples from 46 patients with lung cancer (37 with ADC and 9 with SCLC) who were initially without BM at diagnosis. The cohort consisted predominantly of stage IV cancers (40 patients), with a smaller proportion of stage III (6 patients). The median age at LC diagnosis was 65 years, ranging from 38 to 79, and the majority were male (72%). Patients received various treatments, including PD-L1 inhibitors and standard care chemotherapy (etoposide, paclitaxel, platinum, pemetrexed, or irinotecan) as well as targeted therapies such as osimertinib, erlotinib, afatinib, loratinib, and crizotinib. Over a median follow-up period of nine months, ten patients developed BMs, of which nine were asymptomatic, and one was symptomatic, detected at a median of six months (range: 3–9 months). At the time of BM detection, all these patients exhibited increased predictive values using age, sNfL, and sGFAP, above the established cutoff of 0.4, with a median predicted value of 0.85 (interquartile range [IQR]: 0.7–0.99) (Figure 3A). Interestingly, three patients showed predictive values exceeding the cutoffs 1–3 months before MRI confirmation of BM. However, no MRI was performed when biomarker levels were elevated, leaving it unclear whether BM was present during these elevated biomarker levels. Among the 147 samples from 33 patients who did not develop BMs, most remained stable and below the cutoff level, suggesting the absence of BM development (Figure 3B). However, ten samples from eight patients showed falsely elevated predictive values, resulting in a false positive rate of 7%. Of these, five cases could be attributed to specific factors: two due to chemotherapy-induced peripheral neuropathy and three from other complications (spinal and sacral radiation therapy and subacute stroke). The remaining three patients who showed false positives did not have a clear cause identified.

## 3. Discussion

This study investigated the utility of sNfL and sGFAP as biomarkers for detecting BMs in patients with LC. We demonstrated that sNfL and sGFAP were able to predict the presence of BM accurately with an AUC of 0.837 and 0.775, respectively. Independently, younger age was also found to increase the risk of BM, which is consistent with prior studies [16]. The development of a nomogram, incorporating age and levels of sNfL and sGFAP, achieves high predictive accuracy for BM presence on MRI, evidenced by an AUC of 0.877, demonstrating its effectiveness in clinical assessment. Notably, the model exhibited an impressive NPV of over 93%, suggesting its high efficacy in ruling out BMs during initial LC staging. By leveraging this model, clinicians can more accurately direct targeted MRI evaluations to those most likely to benefit, thus reducing unnecessary testing and optimizing resource allocation in the diagnostic process. Further, a subgroup analysis was conducted among patients with advanced LC to address the possibility of elevated sNfL and sGFAP levels due to disseminated LC rather than specific BM. Multivariable analysis reaffirmed that age, sNfL, and sGFAP are significant predictors of BM. A nomogram incorporating these factors exhibited robust discriminative power (AUC of 0.867), indicating that notable increases in these biomarkers that are likely indicative of BM rather than systemic cancer progression. We also prospectively monitored the trajectory of these biomarkers in patients with advanced LC. Over an average monitoring period of nine months, 93% of samples in patients who did not develop BM remained below the established cutoff level, highlighting their stability in patients without BM. Conversely, all patients who developed BM showed elevated predictive values at detection of BM, with a median predicted value of 0.85, affirming the potential of sNfL and sGFAP as surveillance markers for BM.

Increased sNfL and sGFAP levels in patients with BMs may result from tumor microenvironment changes and host tissue injury [17,18]. In response to injuries caused by cancer cells, astrocytes become activated into a reactive state characterized by high GFAP levels [17]. Although initially reactive astrocytes may attempt to limit metastatic spread by inducing cancer cell death, they ultimately support the establishment and progression of BM through direct interaction and the release of pro-tumorigenic factors [19]. Activated and damaged astrocytes can release GFAP into surrounding tissues or cerebrospinal fluid [20]. Similarly, NfL, a cytoskeletal protein expressed exclusively in neurons, is released from the cerebrospinal fluid into blood proportionally to the extent of brain nerve damage caused by cancer invasion. Moreover, disruptions in the blood–brain barriers may allow GFAP or NfL to cross through CSF to circulation. Previous studies have demonstrated that sNfL and sGFAP levels are associated with the number and size of BM lesions [13,15]. The diagnostic performance of sNfL and sGFAP is better in patients with symptomatic BM than in those without symptoms [15]. Previous findings have demonstrated that elevated sNfL and sGFAP levels at the time of BM diagnosis were associated with poor patient outcomes [13,14], which can be explained by the association between sNfL and sGFAP levels and the extent of BM lesions; thus, the more advanced the BM, the worse the prognosis.

Previously, serum S100B has been suggested as a biomarker for detecting BM in patients with NSCLC [21,22]. However, its effectiveness has yet to be consistently validated in subsequent studies [23], and the low specificity of approximately 43% complicates its clinical utility [21]. Consequently, the role of S100B as a biomarker for BM in NSCLC remains uncertain, necessitating further investigation. A recent study involving patients with NSCLC reported that a predictive model using cathepsin F (CTSF) and fibulin-1 (FBLN1) as diagnostic markers for BM achieved a high sensitivity of 92.6% and a specificity of 87.5% [24], indicating potential but also highlight the need for further validation. Additionally, ongoing research efforts are developing models to predict the future occurrence of BM using proteomics, genes, circulating tumor cells, and circulating tumor DNA. [25,26,27]. Our study contributes to this field by evaluating sNfL and sGFAP as screening markers in a large cohort of mostly asymptomatic patients with LC. This approach offers a practical evaluation of their effectiveness in detecting BMs within a clinical setting. Future research should compare these markers with other emerging liquid-based biomarkers to identify the most effective tools for the early detection of BM in patients with LC.

The strength of this study is grounded in its integration of extensive cross-sectional and longitudinal data encompassing a substantial cohort of patients with LC. Detailed assessments, coupled with adjustments for critical demographic factors like age—which are known to influence sNfL and sGFAP levels—enhance the reliability of these biomarkers in identifying BM. However, limitations include the need for external validation, considering potential variations in biomarker levels influenced by race and ethnic factors. Since these markers are not exclusive to BM and can be affected by various neurological diseases [10,28], there is a likelihood of false-positive outcomes in patients with neurological comorbidities or those who have undergone neurotoxic treatments. Considering that these markers are not specific to BM pathology, they cannot replace MRI but can complement it by aiding in prioritizing MRI scans for those at higher risk of BM. The debate surrounding the routine application of brain MRI for detecting BMs in patients with LC is centered on balancing the benefits of early detection and treatment against considerations such as cost, accessibility, and the relatively low incidence of asymptomatic BMs in the early stages of LC [29,30]. Therefore, personalized MRI screening, informed by these blood biomarkers―which are relatively more accessible and cheaper than MRI―can be expected mitigated the downsides of routine MRI evaluations and enhance their overall benefits.

This study demonstrates that sNfL and sGFAP can serve as effective screening biomarkers for predicting the presence of BM in patients with SCLC or ADC. A nomogram incorporating age and levels of sNfL and sGFAP achieved high predictive accuracy for BM presence on MRI, with an AUC of 0.877 and a high NPV of 93%. The utilization of these biomarkers can reduce the number of unnecessary MRI scans by identifying patients with a low likelihood of BM, while significant increases in these markers can prompt targeted MRI screening, suggesting potential BM development. This strategy optimizes resource allocation and enhances diagnostic efficiency, providing a strategic advantage in the clinical management of patients with LC.

## 4. Materials and Methods

### 4.1. Patients and Study Design

#### 4.1.1. Cohort 1: Retrospective Cross-Sectional Analysis

We identified a consecutive retrospective cohort of newly diagnosed LC, specifically adenocarcinoma [ADC] or SCLC, from 1 July 2020 to 30 June 2022. A total of 768 patients were initially considered. In this study, we included only ADC and SCLC due to their high BM prevalence of 40–50% [1,2,3]. Squamous cell carcinoma was not included as a study subject because of its lower BM prevalence of 8–28% [31,32,33], aiming to enhance the statistical validity of our findings and focus on cancer subtypes where BM detection is most critical. After applying exclusion criteria—age over 80 years (37 patients excluded), recent history of stroke (2 excluded), peripheral nerve disease (2 excluded), and lack of initial brain MRI data (16 excluded)—700 patients were included in the analysis. The Biobank of the National Cancer Center (NCC), Korea, provided all blood samples for these patients and supplied the associated data for analysis. Blood samples were analyzed to measure serum levels of neurofilament light chain (sNfL) and glial fibrillary acidic protein (sGFAP). The presence of BM was determined based on MRI scans performed using a 3.0-Tesla MR scanner (Philips Achieva; Philips Medical Systems, Best, The Netherlands).

#### 4.1.2. Cohort 2: Prospective Observational Analysis

The prospective observational cohort included patients with stage III or IV LC who had no BM at initial diagnosis. These patients were monitored for sNfL and sGFAP levels every three months and underwent brain MRIs every six months, or more frequently if clinically indicated. Monitoring of blood levels ceased upon the confirmation of BM. Patients who received prophylactic cranial irradiation were excluded from this cohort. Of the initial 54 patients enrolled, 8 were excluded from the analysis due to transfer to another hospital (4 patients), death within six months of follow-up (3 patients), and acute stroke (1 patient).

### 4.2. Blood NfL and GFAP Level Assessment

Blood samples for Cohort 1 were collected at the initial lung cancer diagnosis. For Cohort 2, samples were taken every three months post-enrollment. The collected samples were immediately separated by centrifugation and frozen at –80 °C until analysis. The sNfL and sGFAP levels were measured using the Simoa^®^ HD-1 platform (Simoa^®^ NF-light™ Advantage Kit and GFAP* Discovery Kit, respectively; Quanterix, MA, USA) following the manufacturer’s instructions. The intra- and inter-assay coefficients of variation were <10%. All analyses were performed offsite in a blinded manner for sample diagnosis. The blood tests for sNfL and sGFAP take approximately one week to complete. The combined cost for sNfL and sGFAP testing is about USD 200 per sample, which is roughly 30% of the cost of an MRI scan.

### 4.3. Statistical Analysis

Descriptive statistics summarized patient characteristics. Log transformation was applied to address the observed skewness in sNfL and sGFAP levels, mainly due to very high values in patients with advanced BM. This mathematical adjustment stabilizes variance, making the data adhere closely to a normal distribution. The normality of data distribution before and after log transformation was confirmed using the Shapiro–Wilk test. Differences in the distributions of BM between patients with and without metastases in cohort 1 were analyzed using the Wilcoxon rank-sum test (Appendix A). Multivariable regression analyses, adjusted for potential confounders, such as age, sex, body mass index (BMI), diabetes mellitus, hypertension, staging, and cancer histology, were performed to evaluate the association of the biomarkers with the presence of BM. The multivariate model included variables with a *p*-value <0.2 based on the univariate model, and a backward variable selection method with an elimination criterion of 0.05 was used. A nomogram was constructed based on the regression coefficients of the selected model. The performance of the model was assessed by determining the area under the curve (AUC) value, receiver operating characteristics curves, and calibration plots and employing 1000 bootstrap resamples for validation. The Hosmer–Lemeshow test evaluated model’s goodness-of-fit. Statistical significance was set at *p* < 0.05. For Cohort 2, the model developed from Cohort 1 was applied to prospectively evaluate its effectiveness in predicting the onset of BM, with results suggesting the model’s applicability in a clinical setting. Analyses were performed using the R Foundation for Statistical Computing (version 4.1.2, Vienna, Austria).

## Figures and Tables

**Figure 1 ijms-25-06397-f001:**
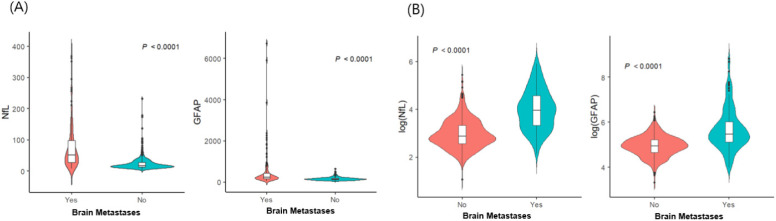
Distribution of serum glial fibrillary acidic protein and serum neurofilament light chain according to the presence of brain metastases (n = 700): (**A**) raw data; (**B**) log-transformed values. Abbreviations: NfL—serum neurofilament light chain; GFAP—glial fibrillary acidic protein.

**Figure 2 ijms-25-06397-f002:**
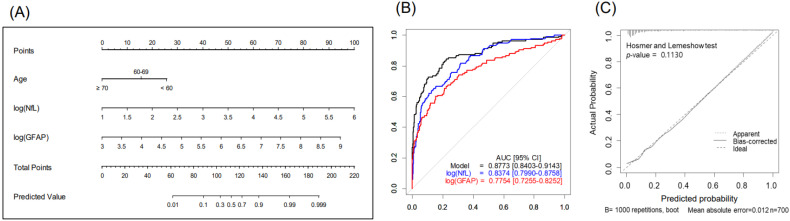
Nomogram for predicting brain metastases in patients with lung cancer (n = 700): (**A**) study design, (**B**) predictive accuracy, and (**C**) calibration curves with Hosmer–Lemeshow test. Abbreviations: NfL—serum neurofilament light chain; GFAP—glial fibrillary acidic protein.

**Figure 3 ijms-25-06397-f003:**
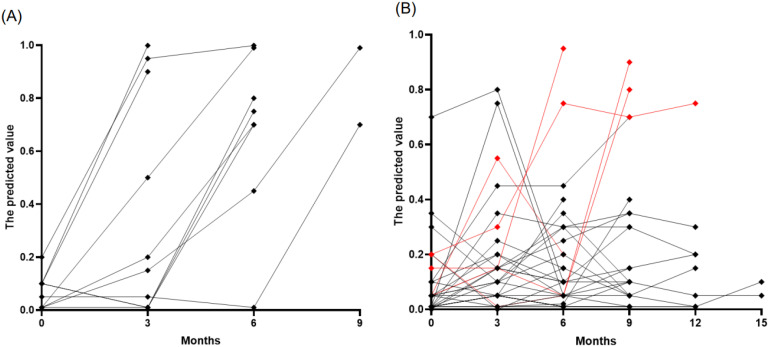
Temporal tracking of predicted brain metastases risk in patients with lung cancer. (**A**) Patients who developed BM (n = 10), highlighting an increase in risk scores at the time of BM diagnosis. (**B**) Patients who did not develop BM (n = 33), with most observations (black points) remaining below the risk threshold of 0.4, indicating a stable, low risk of BM development. The red points denote cases where risk predictions increased due to factors other than BM.

**Table 1 ijms-25-06397-t001:** Baseline characteristics of patients with lung cancer (N = 700).

Parameters		Values
Age		
	Mean ± SD	63.7 ± 9.3
Sex		
	Male	405 (57.9%)
	Female	295 (42.1%)
BMI		
	Mean ± SD	23.9 ± 3.4
Smoking		
	Current smoking	148 (21.1%)
	Previous smoking	196 (28%)
	Never smoked	356 (50.9%)
HTN		
	HTN	130 (18.6%)
	Non-HTN	570 (81.4%)
DM		
	DM	113 (16.1%)
	Non-DM	587 (83.9%)
Histology		
	ADC	626 (89.4%)
	SCC	74 (10.6%)
Staging prior to brain MRI evaluation		
	1	246 (35.1%)
	2	75 (10.7%)
	3	93 (13.3%)
	4	286 (40.9%)
Brain metastasis		
	Yes	135 (19.3%)
	No	565 (80.7%)
Brain metastasis Symptoms (N = 135)		
	No = 0	110 (81.5%)
	Yes = 1	25 (18.5%)
sNfL		
	Median (Min–Max)	20.4 (2.9–369)
sGFAP		
	Median (Min–Max)	151.4 (27.6–6738)

Abbreviations: SD—standard deviation; Min—minimum; Max—maximum; DM—diabetes mellitus; HTN—hypertension; BMI—body mass index; sNfL—serum neurofilament light chain; sGFAP—serum glial fibrillary acidic protein; ADC—adenocarcinoma; SCLC—small cell lung cancer; CI—confidence interval.

**Table 2 ijms-25-06397-t002:** Univariable and multivariable logistic regression to predict brain metastases (N = 700).

		Univariable		Multivariable	
		OR (95% CI)	*p*-Value	OR (95% CI)	*p*-Value
Age	<60 years	1 (ref)	(0.0030)	1 (ref)	(<0.0001)
	60–69 years	0.590 (0.381–0.914)	0.0182	0.402 (0.224–0.722)	0.0023
	≥70 years	0.450 (0.277–0.730)	0.0012	0.098 (0.049–0.198)	<0.0001
Sex					
	Male	1 (ref)			
	Female	0.739 (0.501–1.089)	0.1265		
BMI					
	≤25 kg/m^2^	1 (ref)			
	>25 kg/m^2^	0.490 (0.316–0.762)	0.0015		
Smoking					
	Current smoking	1 (ref)	(0.5584)		
	Previous smoking	0.758 (0.445–1.290)	0.3067		
	Never smoked	0.808 (0.505–1.292)	0.3732		
HTN					
	Non-HTN	1 (ref)			
	HTN	0.448 (0.248–0.808)	0.0076		
DM					
	Non-DM	1 (ref)			
	DM	0.761 (0.442–1.310)	0.3245		
Histology					
	ADC	1 (ref)			
	SCC	2.070 (1.215–3.527)	0.0075		
log(sNfL)					
		6.771 (4.823–9.506)	<0.0001	6.111 (4.059–9.199)	<0.0001
log(sGFAP)					
		7.790 (5.069–11.972)	<0.0001	4.164 (2.472–7.012)	<0.0001

Abbreviations: DM—diabetes mellitus; HTN—hypertension; BMI—body mass index; sNfL—serum neurofilament light chain; sGFAP—serum glial fibrillary acidic protein; ADC—adenocarcinoma; SCLC—small cell lung cancer; CI—confidence interval.

**Table 3 ijms-25-06397-t003:** Points and predictive probabilities for each variable in the nomogram: predicting brain metastases in patients with lung cancer.

**Age**	**Points**	**log(GFAP)**	**Points**	**Total** **Points**	**Predicted** **Value**
<60	20	1	0		
60–69	12	1.2	2		
≥70	0	1.4	5		
log(NfL)	Points	1.6	8	74	0.01
1	0	1.8	10	89	0.05
1.2	3	2	13	95	0.1
1.4	6	2.2	15	99	0.15
1.6	10	2.4	18	102	0.2
1.8	13	2.6	20	105	0.25
2	16	2.8	23	107	0.3
2.2	19	3	25	109	0.35
2.4	22	3.2	28	111	0.4
2.6	25	3.4	30	113	0.45
2.8	29	3.6	32	114	0.5
3	32	3.8	35	116	0.55
3.2	35	4	38	118	0.6
3.4	38	4.2	40	120	0.65
3.6	41	4.4	43	122	0.7
3.8	44	4.6	45	124	0.75
4	48	4.8	48	127	0.8
4.2	51	5	50	130	0.85
4.4	54	5.2	53	134	0.9
4.6	57	5.4	55	140	0.95
4.8	60	5.6	58	155	0.99
5	63	5.8	60	175	0.999
5.2	67	6	62		
5.4	70	6.2	65		
5.6	73	6.4	68		
5.8	76	6.6	70		
6	79	6.8	73		
		7	75		
		7.2	78		
		7.4	80		
		7.6	83		
		7.8	85		
		8	88		
		8.2	90		
		8.4	93		
		8.6	95		
		8.8	98		
		9	100		

Abbreviations: NfL—neurofilament light chain; GFAP—glial fibrillary acidic protein.

## Data Availability

All relevant data are presented in the paper. The raw data generated in this study are available upon request from the corresponding author.

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
