# Peer review of "Blood Neurofilament Light Chain and Glial Fibrillary Acidic Protein as Promising Screening Biomarkers for Brain Metastases in Patients with Lung Cancer"

_ijms, 2024, doi:10.3390/ijms25126397_

Round 1

Reviewer 1 Report

Comments and Suggestions for Authors

The authors present a nice report discussing the use of two serum markers to detect and predict the development of lung cancer brain metastases in a prospective arm of the trial. The data is presented well, however there are a few point which can be improved. The authors should first include a conclusion. Secondly, the authors should discuss the time taken to obtain this lab result and the cost as related to an MRI. The authors state that these tumor markers are possible alternatives to MRIs but I do not foresee how these markers will replace MRIs. These will always be imperative, so it is unclear what role these markers will play in the future. Also, the authors should discuss in more detail how these markers are increased in the bloodstream due to metastases. The pathophysiology should be address briefly in the report. Finally, the authors need to discuss the niche with which this new screening can fit into standard clinical practice, where MRIs are reliable and available for nearly all lung cancer patients. When these are address, this report may be considered for publication. 

Comments on the Quality of English Language

Minor revisions throughout, nothing substantial 

Reviewer 2 Report

Comments and Suggestions for Authors

I read with interest this paper which has the ambitious and delicate aim of assessing whether and how a blood sample can have a prognostic/predictive value in patients with SCLC or NSCLC lung cancer.
A particularly pertinent issue is that of identifying an economically feasible parameter. While magnetic resonance imaging (MRI) is undoubtedly a valuable diagnostic tool, its high cost and variable utilisation, particularly in the absence of clear pathology guidelines, necessitates the search for an alternative, more cost-effective solution.
It is therefore of the utmost importance to emphasise the importance of the basics and the scientific soundness of the proposed approach.
I am uncertain as to the layout of the article. I am uncertain about the materials and methods presented in section 4, as well as the absence of a concluding chapter.
Some comments are offered below. It would be beneficial to include a specific mention in the abstract that the samples in question are blood samples, as well as a brief overview of the most significant findings.
In the Materials and Methods section, it would be beneficial to clarify the rationale behind the decision not to include spinocellular NSCLCs. It is unclear why patients over the age of 80 were excluded from the study. Please provide an itemised breakdown of the costs associated with the materials used for blood samples. The number of BM patients included in the study is too limited (n=135). Has a preliminary analysis of sample adequacy been carried out?
It would be advisable to rewrite the discussion section more thoroughly by comparing the aforementioned data with similar analyses (liquid biopsies and predictive values). Additionally, a conclusion could be included in order to present the argument more effectively. 

Comments on the Quality of English Language

It is recommended that the text be subjected to a linguistic revision

Round 2

Reviewer 2 Report

Comments and Suggestions for Authors

Once the review has been completed, I would like to express my gratitude for the effort that has been made.

Comments on the Quality of English Language

none